# Periprosthetic Fracture After Cementless Revision Total Hip Arthroplasty with a Tapered, Fluted Monobloc Stem: A Retrospective Long-Term Analysis of 121 Cases

**DOI:** 10.3390/jcm14072409

**Published:** 2025-04-01

**Authors:** Oliver E. Bischel, Jörn B. Seeger, Paul M. Böhm

**Affiliations:** 1BG Trauma Center, University of Heidelberg, Ludwig-Guttmann-Str. 13, 67071 Ludwigshafen, Germany; 2Parc Clinic, Am Kaiserberg 2-4, 61231 Bad Nauheim, Germany; joernseeger@gmx.net; 3General Orthopedics, Pläntschweg 25, 81247 Munich, Germany; info@ortho-boehm.de

**Keywords:** modular revision stem, revision THA, periprosthetic fracture, survivorship analysis

## Abstract

**Background**: The use of tapered monobloc stems in revision total hip arthroplasty (RTHA) has shown excellent results, with low implant-dependent failures due to aseptic loosening. Infection is one of the main failure reasons, but further problems, like periprosthetic fractures (PPFs), may endanger the function and duration of the implant in the long run. **Methods**: A consecutive series of 121 cases after femoral RTHA with a monobloc device was retrospectively investigated, and a Kaplan–Meier analysis was performed. The mean follow-up was 13.0 (range: 0.8–23.8) years. **Results**: PPF occurred in six patients during follow-up. The cumulative risk for PPF was 5.2% (95% CI: 1.1–9.4%) after 23.8 years. Female gender was associated with a significantly higher risk compared to male gender (9.1% (95% CI: 2.1–16.1%) after 23.1 years vs. 0% after 23.8 years; log-rank *p* = 0.0034). Patients operated with stems with a length equal to or longer than the calculated median length were also at a significantly higher risk of PPF during follow-up (10.2% (95% CI: 2.4–17.9%) after 23.8 years vs. 0% after 23.1 years; log-rank *p* = 0.0158). Diabetes at the time of index operation also significantly influenced the occurrence of a PPF during follow-up (n = 4 patients with PPF out of 107 without (4.0% (95% CI: 0.2–7.8%) after 23.8 years vs. n = 2 out of 14 with diabetes (15.4% (95% CI: 0–35.0%) after 21.1 years; log-rank *p* = 0.0368). The failure rate with implant removal as an endpoint due to aseptic loosening was 0%, and with infection it was 3.4% (95% CI: 0.1–6.7%), after 23.8 years. **Conclusions**: Although no removal of the implant due to a PPF was necessary, the cumulative risk for PPF after femoral revision with a tapered and fluted monobloc stem was higher in this long-term follow-up series compared to implant failure due to infection or aseptic loosening. Female gender and diabetes was associated with a significantly higher risk of PPF during follow-up. The use of longer stems than necessary is not preventive of PPF, and should be avoided.

## 1. Introduction

The introduction of tapered and fluted revision devices almost 40 years ago improved the outcomes of femoral RTHA dramatically. Compared to other stem designs, like cylindrical implants, tapered and fluted devices are favorable for bridging bony defects, and show lower intraoperative fracture rates, as well as favorable functional results, with less thigh pain [1,2,3]. In addition, rotational stability is prevented as a presupposition for bony integration and long-term durability [2,4]. Mid-to-long-term follow-up studies have underlined the impressiveness of this capability, and have shown that the frequency of aseptic loosening as a failure reason for implant removal is low [5,6,7,8,9].

Nevertheless, age, gender and consecutive osteoporosis may reduce the bony stability of a cementless implant in the long run. After primary THA, a cumulative risk of occurrence of a PPF of a bony integrated implant of up to 10% after more than 20 years is reported [10], but similar investigations into the outcomes of femoral RTHA are missing so far. The amount of RTHAs is rising steadily in industrial countries, due to the demographic development. Consequently, an increasing number of patients presenting with a PPF after cementless femoral RTHA is to be expected. The treatment of PPFs of integrated revision devices in older patients is challenging not only with respect to the surgical procedure, but also due to high complication rates and resulting costs. Therefore, estimation of the cumulative risk of PPF after femoral RTHA in a long-term follow-up period is mandatory in order to provide the adequate infrastructure necessary for the treatment of PPF after RTHA.

## 2. Patients and Methods

### 2.1. Inclusion Criteria, Methods

A consecutive cohort of 121 patients was evaluated retrospectively. In all patients, femoral RTHA was performed using a Wagner SL**^®^** revision stem (1st and 2nd generation, Zimmer, Warsaw, IN, USA). The relevant basic data of the patients and operations are given in Table 1.

Only patients with evidence of a stable integrated stem without loosening or subsidence were included for the survivorship analysis with PPF as a failure criterion. Therefore, a follow-up of at least five weeks was found to be adequate, and corresponds to fracture healing of long bones in adults. An at least partial bony ingrowth of the implant, with sufficient secondary stability, could be assumed at that time. As (low-grade) infection may negatively influence or even prevent bony integration, all infected cases were excluded from the calculation of the cumulative risk of PPF. All cases with subsidence without secondary stabilization and/or instability were also excluded from the study. There was one case with an ingrown stem, but joint instability due to malpositioning of stem and cup, with consecutive RTHA. This case was also excluded (see Table 1). All excluded patients, especially cases with the presence of late infection or loosening, had no PPF until the latest follow-up and/or revision. For comparison reasons, survivorship analysis of the whole cohort was also performed with infection as the endpoint.

Several basic data of the patients, like sex, the presence of diabetes at the date of index operation and BMI, were analyzed. The patients were classified into underweight (BMI < 20 kg/m^2^), normal weight (BMI between 20 and 25 kg/m^2^), overweight (BMI between 25 and 30 kg/m^2^) or obese (BMI > 30 kg/m^2^: 1° between 30 and 35 kg/m^2^; 2° between 35 and 40 kg/m^2^), according to the criteria of the World Health Organisation. Specific implant-related parameters, like stem/reconstruction length, or radiologic factors, like defect classification, according to Paprosky et al. [11], with potential influence on PPF were also investigated and statistically analyzed. The Vancouver classification was used to classify occurring PPFs [12].

Statistical analysis was performed with JMP 10 for Mac (SAS Institute Inc., Cary, NC, USA). A time-to-event analysis was performed using the Kaplan–Meier method, with postoperative PPF and infection as failure criteria. A 95% confidence interval was given to all survivorship data; the *p*-value for comparing survival curves was calculated with the log-rank test.

### 2.2. Surgical Technique and Postoperative Care

The used approaches and operation-related data are listed in Table 1. Auto- and allograft bone was used during operation for defect reconstruction of the femur. Autografts were gained from the cup during reaming in cases with complete exchange of the THA. The use of massive grafts and/or morsellized material is shown in Table 1.

The implant bed of the Wagner SL device with a straight design was prepared by corresponding reamers of the system for all available lengths. The 190 mm stem was used in one hip, the 225 mm length in 16, the 265 mm length in 41, the 305 mm length in 49, the 345 mm length in 12 and the 385 mm stem in two. The stem length was primarily adapted to the underlying bony defect by the surgeon, and according to a preoperative planning.

Two days postoperatively, the patients started with physiotherapy and were mobilized. Partial weight bearing with 20 kg bodyweight was recommended for six weeks. Afterwards, this was followed by a stepwise increase in weight bearing, with 10–20 kg per week, after X-ray control. Active and passive exercises of hip motion were restricted to 60° of flexion for six weeks. Aftercare at specialized rehabilitation units was organized after reaching full weight bearing. Aftercare in a wheelchair was performed for at least six weeks in patients not able to perform partial weight bearing. Weight bearing with half bodyweight was affiliated, and stepwise increased afterwards with 10–20 kg per week.

## 3. Results

### 3.1. Data Collection

Data were available and collected for all revisions, but 54 patients died during follow-up. The mean duration from index operation to death was 9.9 (range: 0.8–21.6) years (Table 2). The data of these patients were included until the most recent follow-up. The mean follow-up period of the cohort is visible in Table 2.

### 3.2. Periprosthetic Fractures

All six PPFs occurred after a low-impact trauma, like stumbling at home (Table 3), at a mean follow-up of 2.9 (0.1–6.8) years postoperatively (Table 2). The Vancouver classification is presented in Table 3, and all fractures were located near the tip of the prostheses. Conservative treatment or open reduction and internal fixation (ORIF) was initiated in three patients each. Conservative treatment was performed by bracing over a period of six weeks, as the fractures were incomplete and not displaced. All fractures healed, and the stems were stable during further follow-up.

### 3.3. Survivorship Analysis and Predictive Factors for PPF

There was no aseptic loosening during follow-up (Table 2). Stem removal due to infection occurred in four patients, and the cumulative risk was 3.4% (95% CI: 0.1–6.7%) after 23.8 years (Table 2). Figure 1 shows the cumulative risk for infection after 23.8 years (Figure 1).

The cumulative risk for PPF was 5.2% (95% CI: 1.1–9.4%) at 23.8 years, and is illustrated in Figure 2 (Figure 2; Table 2). Sex had a significant impact, as all patients presenting with PPF were of female gender (log-rank *p* = 0.0034; Table 2 and Table 3). Figure 3 demonstrates this finding (Figure 3). A significantly higher risk for PPF during follow-up was found in diabetics (n = 2 out of 14) compared to patients without diabetes at the time of index operation (n = 4 out of 107; log-rank *p* = 0.0368; Table 2). The correlation between diabetes and the occurrence of PPF is shown in Figure 4 (Figure 4). The stem length was also found to have a significant influence, as all PPFs occurred in stems equal to or longer than the median used length of 305 mm (log-rank *p* = 0.0158; Table 2). Figure 5 demonstrates this significant correlation (Figure 5).

## 4. Discussion

### 4.1. Background and Rationale

The non-modular Wagner SL revision stem was introduced nearly 40 years ago [13]. Femoral bone defects could be bridged safely with this implant, and primary stability prevented by the tapered and fluted stem design. Studies with an at least mid-term follow-up have shown excellent results [2,4,5,6,7,8,14].

Nevertheless, PPF constituted the reason for 5% of reoperations after RTHA in the Swedish Hip Arthroplasty Register in 2020 [15]. For revision devices, there are limited data available, and investigations over a long-term period are missing.

### 4.2. Cumulative Risk of PPF and Predicting Factors

Although the risk for PPF after RTHA with these cementless devices is lower compared to that with cementless primary THA [10,16], the cumulative risk of 5.2% is higher than the failure rate for aseptic loosening (0%) or infection (3.4%) after 24 years found in this series or in other comparable studies [5,17]. Once stably integrated, aseptic loosening is rare during follow-up, as also shown in other publications, but the presence of PPF may increase, due to the demographic situation and its related problems.

All patients presenting with PPF during follow-up were female. In addition, diabetes was shown to have a negative impact during follow-up, assuming osteoporosis as one risk factor in RTHA, as well as after primary THA [18,19,20,21,22,23]. This finding is unsurprising, as patients undergoing RTHA are usually of older age. Therefore, the presence of PPF after RTHA may be considered as indicative of fractures due to osteoporosis as well, and further diagnostics and therapy independent of the surgical treatment may be necessary.In contrast to primary THA, operative treatment of PPFs with long-stemmed revision stems is more demanding, and restoration of the function of the hip and knee is frequently limited. High costs result not only from acute operative therapy, but also from longer rehab periods or persisting dependency on nursing care. The prevention of osteoporosis and fall-related fractures due to a low-impact trauma, as well as other age-related factors with negative influence on mobility, may be more effective for geriatric humans who are at risk, and for the healthcare system [24,25].

All fractures occurred with stems equal to or longer than the calculated median of all stems implanted in this cohort (Table 2). One explanation could be a lower elastic modulus of the bone through the whole anchoring length of the prosthesis, and a concentration of the stress at the tip of or below the implant. In addition, with cortical thinning below the isthmus, especially with the presence of osteoporosis, the risk of PPF may higher. Similar findings were made after cemented primary THA with Exeter stems and Vancouver type C fractures [26].

In contrast, underlying preoperative bony defects did not correlate with the presence of PPF during follow-up. Too-long stems had been used to bridge the defect in several cases. One explanation may be the limited experience with this implant, as was introduced into surgeries in the late 1980s and 1990s. At that time, little data were available dealing with bony defects bridged by revision devices. In addition, the stem length of the second-generation Wagner SL**^®^** device could only be estimated by marks on the reamer, and had not been verified by testing devices intraoperatively. Both of these findings may explain the difference between the underlying defect and used implant length.

Consequently, the use of longer implants than necessary in RTHA may not be preventive for PPF compared to cemented fixation in primary THA. In addition, the removal of integrated stems and/or adequate fixation by plates in the case of ORIF may be more difficult with longer devices in situ.

All fractures were of type B1 or C, according to the Vancouver classification, and we did not investigate any B2 or 3 situations revealing a loosened stem. This monobloc revision device offers enough bony integration, and therefore secondary stability, despite an initial situation with a highly deficient bone stock and/or even osteoporitic bone [2,4,5,6,7,8,14].

### 4.3. Limitations

There may be limitations due to the study protocol, as it is a retrospective study without a control group. Nevertheless, a high number of patients could be investigated during a long-term duration. Application of Kaplan–Meier analysis was the method of choice, and it expressed deviating conclusions. Investigation of osteoporosis and further diagnostics were not performed routinely during follow-up, but may also help to identify patients who are at risk. Nevertheless, there are clear advantages in the use of tapered and fluted stems in femoral RTHA when weighing up the pros and cons.

## 5. Conclusions

Although no implant needed removal due to PPF in this long-term follow-up series, the cumulative risk of PPF during follow-up, after femoral revision THA with a cementless, tapered monobloc device, was high compared to the risk of infection or aseptic loosening. Gender and diabetes are influencing factors for PPF during follow-up. The implant used should be as short as possible to prevent complications and maintain surgical options. Screening and therapy for osteoporosis, as well as prevention of falls, may help to reduce PPF after RTHA.

## Figures and Tables

**Figure 1 jcm-14-02409-f001:**
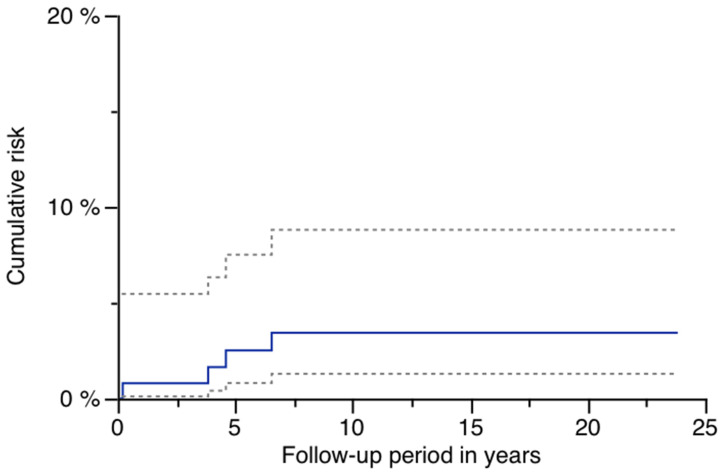
Risk of infection: blue line with 95% CI (dashed lines).

**Figure 2 jcm-14-02409-f002:**
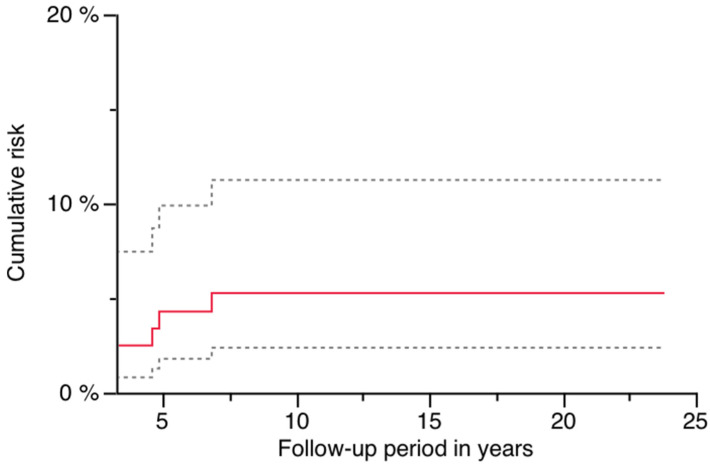
Risk of PPF: red line with 95% CI (dashed lines).

**Figure 3 jcm-14-02409-f003:**
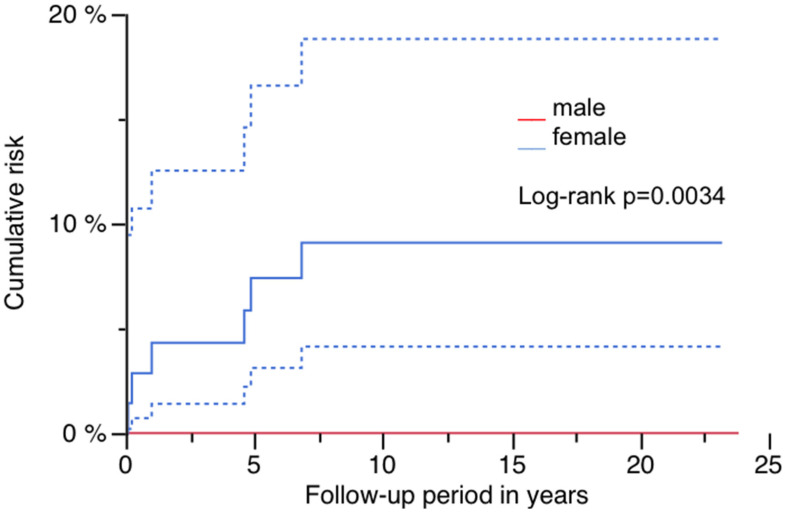
Risk of PPF by gender: red line male and blue line female with 95% CI (blue dashed lines).

**Figure 4 jcm-14-02409-f004:**
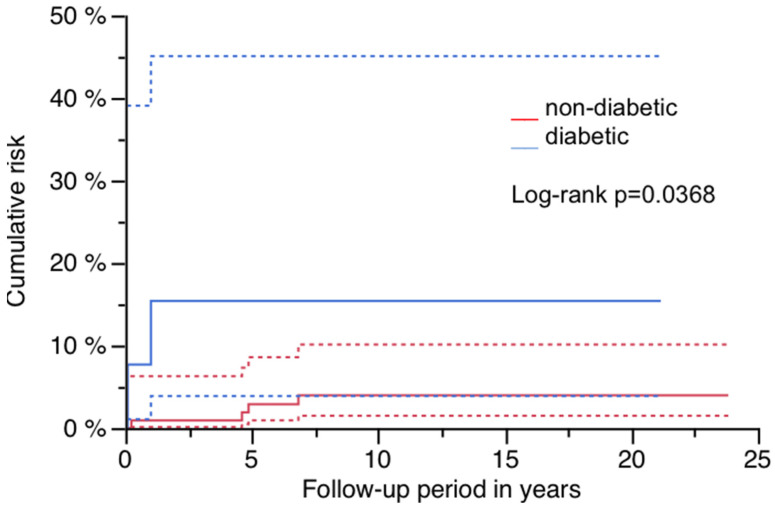
Risk for PPF by diabetes status: red line non-diabetics and blue line diabetics, each with 95% CI (dashed lines).

**Figure 5 jcm-14-02409-f005:**
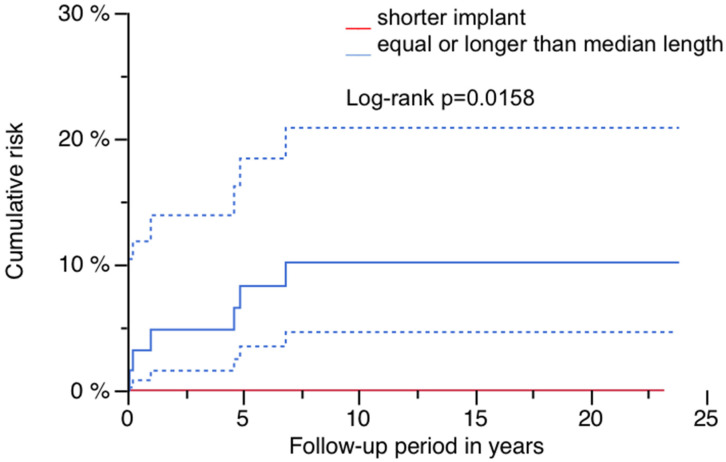
Risk of PPF by implant length: red line shorter implants and blue line equal or longer implants than median length with 95% CI (dashed lines).

**Table 1 jcm-14-02409-t001:** Basic data of patients at index operation.

	Wagner SL
Original cohort:	129
Patients excluded due to following reasons:	
Infection	4
Aseptic loosening	0
Subsidence within six weeks postoperatively and consecutive instability	1
Instability	2 ^§^
Intraoperative PPF ^#^	1
Lost to follow-up	0
Included no. of patients	121
Implanted between (year)	1989–1996
Indication:	
Aseptic loosening	91
Septic two-stage revision	3
Septic one-stage revision	15
Periprosthetic fracture	12
Instability	0
Surgeons involved	7
Sex (w/m)	70/51
Operated side (r/l)	60/61
Mean age at surgery (range) in years	65.28 (36.66–86.26)
Surgical approach:	
Transfemoral	59
Transtrochanteric	3
Hardinge/transgluteal	42
Posterior	16
Anterior	1
Preoperative bone defect (Paprosky)	
Grade 1	24
Grade 2	72
Grade 3A	5
Grade 3B	14
Grade 4	6
Bone transplant at femur:	
Total	47
Autogeneous	10
Allogeneous	35
Both	2
Morsellized	38
Bulk/strut graft	1
Both	8

^#^ and insufficient osteosynthesis; ^§^ one patient with initial instability due to malpositioning of cup and stem; one patient with instability due to cup loosening over 12.9 y after initial index operation using a tapered revision device, with additional transfemoral exchange of stem.

**Table 2 jcm-14-02409-t002:** Results.

	Wagner SL
Mean follow-up (ys.) *	13.0 (0.8–23.8)
Death during follow-up	N = 54
Follow-up of patients who died	9.9 (0.8–21.6)
Mean reconstruction length (range)/median in mm	285.2 (190–385)/305
Mean stem diameter (range)/median in mm	16.6 (14–22)/16
BMI in kg/m^2^	26.4 (17.0–36.3)
Obesity 2°	2
Obesity 1°	22
Overweight	46
Normal weight	46
Underweight	5
PPF during follow-up	N = 6
Follow-up until PPF (ys.)	2.9 (0.1–6.8)
Overall risk (95% CI) of aseptic loosening after years in %	0 after 23.8
Overall risk (95% CI) of infection after years in %	3.4 (0.1–6.7) after 23.8
Overall risk (95% CI) of PPF after years in %	5.2 (1.1–9.4) after 23.8
Risk of PPF and sex (female vs. male) after years	9.1 (2.1–16.1) after 23.1 vs. 0 after 23.8 Log-rank *p* = 0.0334
Risk of PPF and stem length (equal to or longer than median length vs. shorter devices) after years in %	10.2 (2.4–17.9) after 23.8 vs. 0 after 23.1 Log-rank *p* = 0.0158
Risk of PPF and diabetes (diabetic vs. non-diabetic) at time of index operation after years in %	15.4 (0–35.0) after 21.1 vs. 3.8 (0.2–7.8) after 23.8 Log-rank *p* = 0.0368

* failures due to PPF included.

**Table 3 jcm-14-02409-t003:** Periprosthetic fractures.

Parameter	Patients: Age at Surgery (ys.), Gender					
	40, f.	57, f.	75, f.	84, f.	72, f.	79, f.
Indication ^#^	AL	AL	AL	PPF	AL	AL
Bone defect *	2A	2A	3B	3B	3B	3B
Approach ^§^	TG	TG	*p*	TG	*p*	TF
BMI (kg/m^2^)	20	35	19	20	29	22
Diabetes (y/n)	N	Y	N	N	N	Y
PPF postop. (ys.)	6.8	0.1	4.6	0.2	4.9	1.0
Vancouver classification	C	B1	B1	B1	C	C
Conservative (C) vs. operative therapy (O)	O (ORIF by plate)	O (ORIF by plate)	C	C	O (ORIF by plate)	C
Stem/recon-struction length/diameter (mm)	305/16	305/17	305/20	305/21	305/14	305/14

^#^ aseptic loosening (AL); periprosthetic fracture (PPF); * Paprosky classification system; ^§^ transgluteal-lateral/Hardinge (TG); transfemoral (TF); posterior (*p*).

## Data Availability

The original contributions presented in this study are included in the article. Further inquiries can be directed to the corresponding author.

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
