# Peer review of "Periprosthetic Fracture After Cementless Revision Total Hip Arthroplasty with a Tapered, Fluted Monobloc Stem: A Retrospective Long-Term Analysis of 121 Cases"

_jcm, 2025, doi:10.3390/jcm14072409_

Round 1
Reviewer 1 Report
Comments and Suggestions for Authors
This study was very interesting, and I gained a lot of insights from reading it. I have a few comments for improvement that I hope will be helpful.
- What criteria were used to define Overweight, Normweighted, and Underweight? If the WHO classification was used, please specify this in the Methods section for clarity.
- Ensure consistency in the capitalization of “Y” in Table 3 to maintain uniform formatting.
- The classification of periprosthetic fractures using the Vancouver system should be clear and mutually exclusive. The use of “B1/C” is ambiguous and does not align with the standard Vancouver classification. Each fracture should be categorized as either B1 or C, based on implant stability and fracture location. Please clarify and assign a definitive category to each case.
- Additionally, the term “Condylar” is unnecessary in this context. Simply using “Plate” would be clearer and help avoid confusion. Please revise accordingly.
- This study identifies DM and female sex as risk factors for PPF. Have these factors been previously reported as significant risk factors for PPF in the literature? If so, referencing relevant studies in the Discussion section would strengthen the argument.
- The study found that longer stems were associated with a higher risk of PPF. Could this be due to stress concentration at the distal end of the stem, particularly in Vancouver type C cases? Additionally, as the femoral cortex becomes thinner distally, the risk of fracture may increase. It would be beneficial to briefly discuss this hypothesis in the Discussion section.
- Interestingly, this study did not observe any Vancouver type B2/B3 fractures. While this may be coincidental, it could also suggest that the tapered monobloc stem provides strong osseointegration and mechanical stability. If the authors find this interpretation reasonable, they may consider adding a brief discussion on this point, possibly comparing it with previous studies on tapered monobloc stem integration.
Author Response
Dear reviewer,
thank you very much for your comments. We also think that your remarks will help to improve the statements of the paper. Please find a point-by-point response below which hopefully addresses the topics you mentioned.
1. What criteria were used to define Overweight, Normweighted, and Underweight? If the WHO classification was used, please specify this in the Methods section for clarity.
Yes, the classification was made according to the WHO system. The specification was added in the methods section and in table 2.
2. Ensure consistency in the capitalization of “Y” in Table 3 to maintain uniform formatting.
The formatting was adapted and errors corrected
3. The classification of periprosthetic fractures using the Vancouver system should be clear and mutually exclusive. The use of “B1/C” is ambiguous and does not align with the standard Vancouver classification. Each fracture should be categorized as either B1 or C, based on implant stability and fracture location. Please clarify and assign a definitive category to each case.
We have categorized the fractures either in B1 or C. Fractures beginning right at the tip were classified as C. From a biomechanical point of view there is little difference with a fracture line little proximally or distally of the tip especially when performing ORIF by plate. Therefore, the classification was made be B1/C.
4. Additionally, the term “Condylar” is unnecessary in this context. Simply using “Plate” would be clearer and help avoid confusion. Please revise accordingly.
We have deleted additional specifications.
5. This study identifies DM and female sex as risk factors for PPF. Have these factors been previously reported as significant risk factors for PPF in the literature? If so, referencing relevant studies in the Discussion section would strengthen the argument.
In addition to the three sources we mentioned in the discussion chapter, further publications were attached to strengthen this finding.
6. The study found that longer stems were associated with a higher risk of PPF. Could this be due to stress concentration at the distal end of the stem, particularly in Vancouver type C cases? Additionally, as the femoral cortex becomes thinner distally, the risk of fracture may increase. It would be beneficial to briefly discuss this hypothesis in the Discussion section.
We have discussed this and added one literature with identic findings after cemented primary THA and occurring type C fractures.
7. Interestingly, this study did not observe any Vancouver type B2/B3 fractures. While this may be coincidental, it could also suggest that the tapered monobloc stem provides strong osseointegration and mechanical stability. If the authors find this interpretation reasonable, they may consider adding a brief discussion on this point, possibly comparing it with previous studies on tapered monobloc stem integration.
We have mentioned this in the second section of the discussion chapter 4.2. (‘Once stable integrated, aseptic loosening is rare during follow-up… ‘) and gave two literature sources. We again added a brief remark at the end of the discussion chapter.
Kind regards

Reviewer 2 Report
Comments and Suggestions for Authors
Dear Authors,
Thank you for submitting this insightful manuscript on periprosthetic fractures (PPF) following revision total hip arthroplasty (RTHA) with a tapered, fluted monobloc stem. Your work addresses an important question: What is the long-term risk of PPF after RTHA, and what factors influence it? This is both original and highly relevant to orthopedic surgery, as it tackles a gap in the literature—namely, the scarcity of long-term data on PPF in RTHA compared to primary THA. By providing a 23.8-year follow-up, your study adds valuable evidence to the field, surpassing the shorter-term outcomes typically reported and highlighting PPF as a more frequent concern than aseptic loosening or infection in this context.
The study’s contribution is significant, offering practical insights into risk factors like female gender, diabetes, and stem length that can guide surgical decision-making. Compared to existing literature, it uniquely emphasizes the long-term perspective and challenges assumptions about using longer stems prophylactically, which is a novel and actionable finding.
To enhance the manuscript, a few gentle suggestions:
- Methodology: The methods are robust, but clarifying why five weeks was chosen as the stability threshold (e.g., citing supporting evidence) and how stem length was selected (e.g., surgeon preference or defect-driven) would strengthen transparency.
- Conclusions: Your conclusions align well with the evidence, clearly linking gender, diabetes, and stem length to PPF risk, and they effectively address the main question. Adding a brief clinical recommendation (e.g., screening for osteoporosis) could further tie the findings to practice.
- References: The references are appropriate and well-chosen, supporting your claims effectively.
- Tables and Figures: The tables are detailed and informative—great work! For Table 3, clarifying the “B1/C” Vancouver classifications (e.g., mixed types or uncertainty?) would avoid confusion. Adding short text descriptions for Figures 1–5 (e.g., “Figure 3 shows higher PPF risk in females…”) would improve accessibility.
Overall, this is a strong and compelling study. With minor refinements for clarity and context, it will make an excellent contribution to the field. Thank you for your thorough work!
Best regards.
Comments on the Quality of English LanguageThe English could be improved to more clearly express the research.
Author Response
Dear reviewer,
thank you very much for your comments. In our opinion, the topic of the paper deals with a rising problem and orthopedic surgeons performing adult reconstructive surgery will be confronted with these patients in the future.
We also think that your remarks will help to improve the statements of the paper. Please find a point-by-point response below which hopefully addresses the topics you mentioned.
- Methodology: The methods are robust, but clarifying why five weeks was chosen as the stability threshold (e.g., citing supporting evidence) and how stem length was selected (e.g., surgeon preference or defect-driven) would strengthen transparency.
This threshold was based on radiological findings and a migration analysis according to Callaghan et al.. As described in the second section in ‘methods’ only stable and integrated stems were topic of this investigation. Patients with relevant subsidence were excluded assuming an at least beginning secondary stabilization or mechanical stability within this time of the remaining patients. In addition, bony integration corresponds more or less to fracture healing of long bones in adults.
We have added two sentences to this remark.
In the late 80’s and 90’s of the last century, there was little data available dealing with bony defects and its bridging by revision devices in revision total hip arthroplasty. In addition, there was no testing option during operation and length of the stem was assumed by marks on the reamers. Consecutively, implantation of longer stems resulted. This may be an explanation of this inverse correlation between the underlying bone defect preoperatively and the used implant.
We have discussed this and added this accordingly.
- Conclusions: Your conclusions align well with the evidence, clearly linking gender, diabetes, and stem length to PPF risk, and they effectively address the main question. Adding a brief clinical recommendation (e.g., screening for osteoporosis) could further tie the findings to practice.
Thank you for this remark. Diagnosis and therapy of osteoporosis as well as prevention of fall are mandatory to reduce PPFs after primary or revision THA. We have added this recommendation in ‘conclusions’.
- References: The references are appropriate and well-chosen, supporting your claims effectively.
According to suggestions of other reviewers, we have added further references to the paper.
- Tables and Figures: The tables are detailed and informative—great work! For Table 3, clarifying the “B1/C” Vancouver classifications (e.g., mixed types or uncertainty?) would avoid confusion. Adding short text descriptions for Figures 1–5 (e.g., “Figure 3 shows higher PPF risk in females…”) would improve accessibility.
We corrected this and all fractures were clearly classified. From a biomechanical point of view, there is little difference in the treatment and outcome of a fracture beginning shortly above the tip (B2) or subapical of the stem. For those fractures, B2/C was classified.
Short descriptions were added to each figure.
Kind regards